# Communication-efficient Quantum Federated Learning Optimization for Multi-Center Healthcare Data

Amandeep Singh Bhatia, Mandeep Kaur Saggi, and Sabre Kais

*Abstract*—In the healthcare sector, the scarcity of data and privacy concerns present formidable challenges to the widespread adoption of machine learning. In the present-day scenario, Federated Learning (FL) emerges as a pivotal solution, fostering the rapid evolution of distributed machine learning paradigms while adeptly addressing the problem of data governance and privacy. It allows distributed clients to collaboratively train a global model by synchronizing their local updates without sharing private data. In recent years, federated learning and quantum computing have individually shown great promise to revolutionize various sectors, including healthcare, finance, and manufacturing, where privacy protection is paramount. In this article, we propose a communication-efficient Quantum Federated Learning (QFL) framework based on a variational circuit that enables clients to efficiently train and transmit quantum model parameters, thereby reducing communication rounds significantly and enhancing QFL performance using quantum natural gradient descent (QNGD) optimization. This paper demonstrates the feasibility of a QFL framework for predicting the presence of coronary heart disease, diagnosing whether a patient is suffering from diabetes or not, and differentiating malignant and benign cancer by distributing the UCI datasets unbalanced among healthcare institutions. The proposed framework has the potential to incorporate privacy, security, and the expedited processing of distributed data. QNGD outperformed classical GD by reducing communication rounds by a range of 5% to 60%. In addition to reducing the communication rounds by optimizing the QFL training algorithm and achieving quicker convergence, it also determines the important features regardless of the data imbalance among the clients.

*Index Terms*—quantum machine learning, federated learning, healthcare, variational quantum circuit, quantum optimization

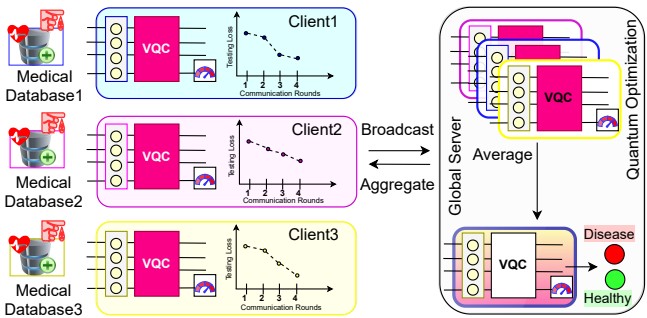

Fig. 1. **An overview of FedVQC** Initially, the global server broadcasts weight parameters to the hospitals/clients. Subsequently, clients conduct training on the received quantum model, utilizing their local medical repository, and then transmit the parameters to the server. In FedVQC, each client employs a quantum natural gradient descent optimization with its local data distribution to refine the parameters of the quantum circuit instead of classical optimization.

## I. INTRODUCTION

IN today's context, Federated Learning (FL) [1] has emerged as a remedy for the healthcare sector, enabling training across multiple hospitals without sharing private data. It addresses the critical need for collaborative model training while respecting patient privacy and data security concerns. In a federated learning framework, the central/global server initializes the machine learning model and distributes it to multiple hospitals, each acting as a client. Subsequently, each client conducts training using its local medical datasets and transmits the refined model updates back to the central hub. Upon receiving updates from all clients, the central server undertakes an aggregation process, thereby updating the status of the global model. The updated model is then redistributed to all clients for subsequent rounds of training. This iterative process continues until the global model converges to a state, where it accurately represents the knowledge learned from the diverse datasets across all hospitals. However, the heterogeneous nature of client datasets in federated learning settings often results in sluggish and erratic convergence, impeding the efficiency of the learning process [2].

In parallel, Quantum Machine Learning (QML) [3] has emerged as an enticing application of quantum technology and received significant attention from academic institutions and research communities alike. It uses the superposition, entanglement, parallelism and other characteristics of quantum computing to improve the performance of machine learning tasks. Till now, several approaches have been proposed and shown great potential to revolutionize various sectors, including healthcare, finance, chemistry, cybersecurity, optimization, and many more [4], [5]. However, the current limitations of quantum computers, including noise and limited scalability, hinder the feasibility of many prominent quantum algorithms for handling practical problems [6], [7].

Currently, Variational Quantum Algorithms (VQAs) are primarily designed for implementation on quantum computers during the noisy intermediate-scale quantum (NISQ) era [8], [9]. To address these constraints, VQAs use a classical optimizer to refine the parameters of a parametrized quantum

Amandeep Singh Bhatia, Mandeep Kaur Saggi, and Sabre Kais are with the Department of Electrical and Computer Engineering, North Carolina State University, Raleigh, North Carolina 27606, USA and Department of Chemistry, Purdue University, West Lafayette, Indiana 47907, USA (email: drasinghbhatia@gmail.com, drmandeepsaggi@gmail.com, skais@ncsu.edu)

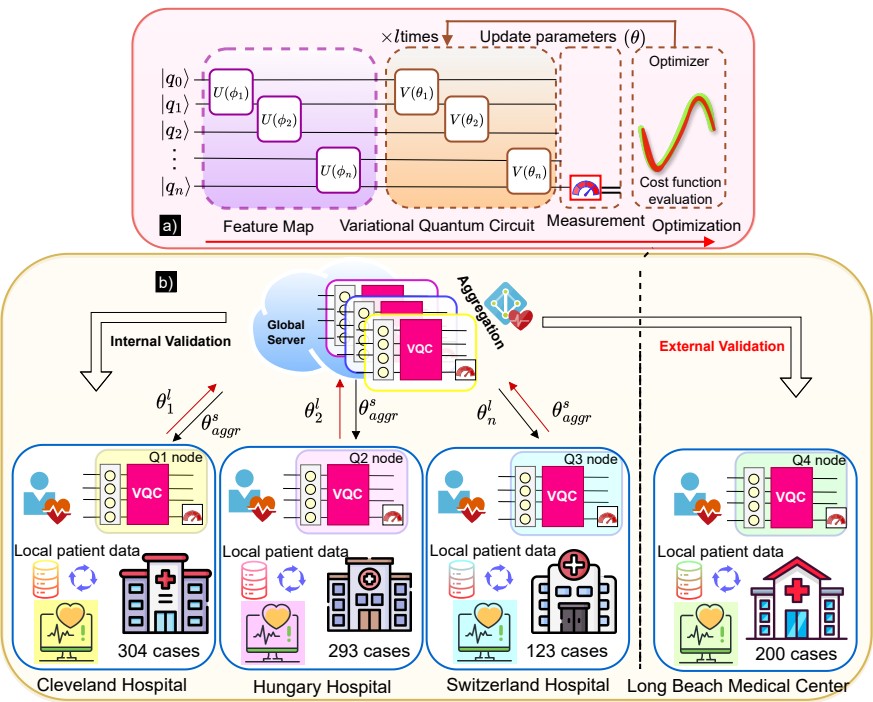

Fig. 2. **Overview of a variational quantum classifier in federated settings for heart disease detection**. It includes (a) A variational quantum circuit comprising a feature map for classical-to-quantum data encoding, a variational circuit, and a measurement process for classifying heart disease vs. no heart disease, followed by classical optimization. (b) A heart disease example. Quantum federated learning framework for heart disease, where n-quantum models are trained on local datasets. An aggregator (central server) collects and integrates local updates to generate a global model.

circuit during training. A VQA implementation consists of three steps (a) Encoding feature map: The process of encoding a classical input data $x$ into a quantum state $|f(\theta, x)\rangle$ using a feature map circuit. It can be implemented in different ways such as basis, angle, amplitude, and arbitrary encoding methods. (b) Variational quantum circuit: The encoded data is processed through the variational quantum circuit (VQC), which serves as an ansatz whose parameters are trained using various optimization methods. (c) Measurement: It involves computing the gradients of a quantum circuit by approximating the expectation value of an observable for $\theta$. An overview of a variational quantum classifier in federated settings is depicted in Fig 1. An overview of a variational quantum classifier is shown in Fig 2(a). An overview of a variational quantum classifier in federated settings for heart disease detection is shown in Fig 2(b).

The iterative optimization process involves adjusting the parameters ($\theta$) of a variational quantum circuit V($\theta$) using a classical or quantum optimization method, to minimize the associated cost function. Variational parameterized circuit architectures are recognized for their capability to produce feature map encodings that lead to readily trainable loss functions [8], [9]. The choice of optimizer plays a crucial role in achieving effective generalization. Although, the trainability of variational quantum circuits suffers from the curse of "barren" plateaus, extensively discussed in recent literature [10]. It is the landscape where the gradient becomes exponentially small in the number of qubits. Nonetheless, even with the current con-

straints of quantum hardware capabilities, variational quantum algorithms remain at the forefront as the preferred approach for gaining an advantage.

In today's context, researchers have only produced a few related works on quantum federated machine learning. Some notable examples of quantum machine learning algorithms in federated settings include quantum neural networks using pre-trained classical models [11], quantum neural networks [12], variational quantum circuits [13], quanvolutional neural networks [14], quantum tensor networks [19], variational quantum circuits in manufacturing [20], and the application of quantum federated machine learning to handle privacy sensitive clinical data [15].

## A. Current challenges

In recent years, there's been a growing reluctance among healthcare organizations to share data or participate in big data communities. This hesitance stems from various factors, including concerns about healthcare data breaches, regulatory compliance, and the potential misuse of sensitive information [16]. Another significant challenge to leveraging data analytics is the sheer volume of healthcare data required to train and validate machine learning models effectively. Particularly deep learning architectures, with millions or even billions of parameters, require vast amounts of high-quality data, which can be challenging to obtain and maintain, especially while adhering to strict privacy regulations.

A challenge in FL is the high communication cost of exchanging weight updates between the global server and the clients. Heterogeneous client datasets lead to slow and unstable convergence, hindering the effectiveness of the FL process. However, while implementing multiple local updates before global aggregation can significantly reduce communication costs, it can increase the computational load on the client side.

*B. Motivation*

Motivated by the versatility and effectiveness of variational quantum circuits, a quantum federated model is introduced to address the reluctance of healthcare entities to share data directly. This model not only prioritizes data privacy but also fosters collaborative analysis and insights, thereby presenting a holistic solution to the challenges faced in the healthcare sector. The primary aim is twofold: to harness the untapped advantages of quantum machine learning and federated learning within the healthcare sector and to alleviate the high communication costs incurred by the frequent transmissions between the FL global server and clients using quantum optimization.

*C. Main contributions*

The main contributions of this paper can be summarized as follows:

- We propose a quantum federated learning framework (QFL) based on variational quantum algorithms and demonstrate its learning capability on popular UCI medical machine learning datasets, and validate our approach and its adaptability to heterogeneous data. Furthermore, we conduct validation of the quantum global model (FedVQC) using both local and external datasets.
- We opt for a quantum variant of gradient descent optimizer instead of classical optimization to tune the parameters of a variational quantum circuit. Through experiments, we demonstrate the efficiency of our QFL framework utilizing quantum optimization in significantly reducing communication costs compared to the classical optimization method.

## II. QUANTUM FEDERATED LEARNING OPTIMIZATION

In this section, we will explore how to harness the power of quantum federated learning through a combination of quantum and classical optimization techniques. Our aim is to enable collaborative training with improved accuracy and enhanced privacy within healthcare institutions. To achieve this, we concentrate on optimizing Variational Quantum Circuits (VQCs) for diagnosing heart disease, diabetes, and breast cancer. This involves leveraging pure quantum AI-driven analysis within the federated learning environment.

We present a QFL algorithm that is an effective solution for collaborative training at specific hospitals, leveraging quantum computing to achieve superior model performance while preserving data privacy. The first step is to encode classical data into a quantum state. To efficiently simulate quantum circuits, we implemented both angle/qubit and amplitude encoding

strategies for quantum state preparation. In angle encoding, the $n$ qubits are utilized to depict $n$-dimensional data, with the input data features encoded into the rotation angles of the qubits. Any $n$-dimensional feature vector $x = [x_1, x_2, ..., x_n]$ is encoded into a quantum state as:

$$|x_i\rangle = \bigotimes_{i=1}^{n} \cos(x_i)|0\rangle + \sin(x_i)|1\rangle \tag{1}$$

Amplitude encoding stores the normalized classical $N$-dimensional input data ($N = 2^n$) in the amplitudes of an $n$-qubit quantum state $|\psi\rangle$ as $|\psi\rangle = \frac{1}{\|x\|} \sum_{j=1}^{N} x_j |j\rangle$.

In amplitude encoding, the number of qubits needed to represent $n$ classical bits is logarithmic, typically $O(\log n)$. After encoding, the next step involves applying a quantum circuit $V(\theta)$ with limited depth ($l$) to the feature state $|\psi\rangle$, depending on the chosen parameterization for the gates and the number of layers. After angle encoding, each layer of VQC consists of single-qubit gates ($R_z, R_y, R_z$) on each qubit, followed by a linear arrangement of control $Z$ gates. After performing amplitude encoding, we utilize a dense quantum circuit, where each layer consists of single qubit rotations and entangle them with control Not gates.

The goal is to determine a sequence of gates that collectively yield the final state $|\psi_o\rangle$. The quantum/classical optimizer manages these parameters throughout training, aiming to minimize a specified loss function. Following this, postprocessing is employed to calculate the circuit's expectation value, ultimately yielding the conclusive outcome of the classifier. The objective is to identify the optimal classifying circuit $V(\theta)$ that effectively distinguishes the dataset with distinct labels.

Suppose there exists clients/hospitals ($H$) and each hospital ($h \in H$) has its own medical data containing $n_d$ samples, denoted state $\{|\psi_i^d\rangle\}_{i=1}^{n_d}$. Thus, the medical data at each hospital consists of unique ids, represented as

$$\rho_d = \frac{1}{n_d} \sum_{i=1}^{n_d} |\psi_i^h\rangle \langle\psi_i^h| \tag{2}$$

Initially, a variational quantum circuit (VQC) is employed for local training. For client $h \in H$, the vector of trainable parameters ($\theta$) is represented by $\overrightarrow{\theta}^h = (\theta_1, \theta_2, ..., \theta_{N-1}, \theta_N)^\intercal$.

A cost function ($C_f$) is defined as the square of trace distance ($T$) between final $|\psi_o o^h\rangle$, and initial state $|\psi\rangle$, which is calculated as

$$C_f(\overrightarrow{\theta}^h) = Tr[O_f V(\overrightarrow{\theta}^h) |\psi_o^h\rangle \langle\psi_o^h| V(\overrightarrow{\theta}^h)^\dagger] \tag{3}$$

where $O_f = 1 - |0\rangle\langle 0|$. It is identical to $C_f = T(|\psi_o^h\rangle\langle\psi|)^2$.

*A. Local model optimization*

Our aim in client-side optimization is twofold: to minimize the divergence of clients from the global model and to reduce communication costs. To achieve this, we employed both classical gradient descent (CGD) and quantum natural gradient descent (QNGD) optimization techniques to refine the parameters of the local VQCs at each communication round. Classical optimizer aims to minimize a cost function $V(\theta)$ by

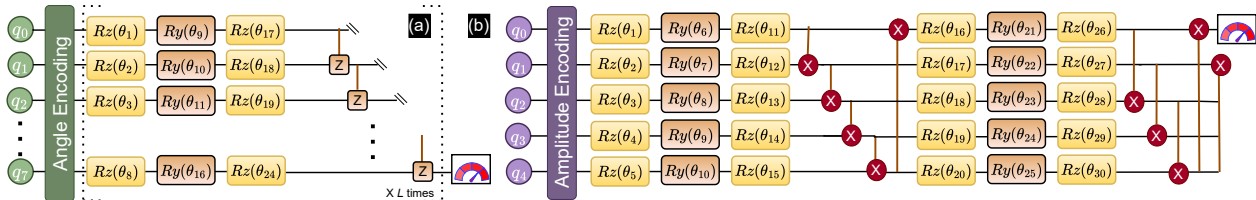

Fig. 3. **Typical structure of a quantum circuit with two encodings** (a) Angle encoding: A 8-qubit variational quantum circuit is structured with angle encoding using Ry, Rz, and Ry gates and linear entanglement. (b) Amplitude encoding: A 5-qubit circuit is depicted, utilizing the Ry, Rz, and Ry gates along with CNOT gates (in red), with the measurement taken from qubit 0. In dark purple, variables are embedded into the amplitudes of a quantum state. In yellow, trainable generic rotational gates are to be optimized during the training phase.

iteratively updating the parameters $\theta$ in the opposite direction of the gradient of the cost function with respect to $\theta$. However, this approach often encounters challenges when dealing with complex, high-dimensional optimization landscapes. The update rule for classical gradient descent is represented as $\theta_{r+1} = \theta_r - \eta \nabla V(\theta_r)$, where $\theta_r$ is the parameter vector at round $r$ m $\eta$ is the learning rate, determining the step size for each update, and $\nabla V(\theta_t)$ is the gradient of the cost function $V(\theta)$ with respect to $\theta$ evaluated at $\theta_r$. While CGD guarantees convergence to a local minimum for convex functions, its performance in non-convex optimization landscapes can be sensitive to the choice of $\eta$ and the initialization of parameters.

Quantum Natural Gradient Descent (QNGD) [18] is an optimization technique that operates on the complex projective space and utilizes the Quantum Fisher Information or Fubini-Study metric to improve convergence speed. The quantum state space features an invariant metric tensor referred to as the Fubini–Study metric tensor, which can be used to develop a quantum version of a natural gradient descent. In QNGD, the cost function and the gradient are transformed in a way that preserves the geometry of the parameter space. The update rule for quantum natural gradient descent is represented as:

$$\theta_{r+1} = \theta_r - \eta F^{-1}(\theta_r)\nabla V(\theta_r) \quad (4)$$

where $\theta_r$ represents the parameter vector at round $r$, $\eta$ is the learning rate, $F(\theta_t)$ denotes the quantum Fisher information matrix evaluated at $\theta_r$, $\nabla V(\theta_r)$ is the gradient of the cost function $V(\theta)$ with respect to $\theta$ at $\theta_r$, and $F^{-1}(\theta_r)$ signifies the inverse of the quantum Fisher information matrix.

QNGD optimizer calculates the block-diagonal metric tensor during each optimization step, requiring $n$ quantum evaluations. The analytic gradient of the objective function is determined $\nabla V(\theta)$ using the parameter shift rule by introducing a small parameter shift $s$ in $\theta$ values, its gradients can be calculated as $\nabla V(\theta_r) = V(\theta + s) - V(\theta - s)$.

### B. Global Aggregation

After performing the local training on the client side, the vector of trainable parameters $(\overrightarrow{\theta}^h)$ is uploaded to the global server, while the local data stays at each hospital. Finally, an aggregation is performed of all local updates $(\theta_r^h)$ and a global

server uses a cost function to minimize the gap between the predictions and actual values.

$$\theta_r^s \leftarrow \theta_r^h - \eta_s \sum_{h=1}^{H} \frac{1}{n_d} \theta_{r+1}^h \quad (5)$$

where $n_d$ is the number of samples available with the client and $\eta_s$ is the learning rate of a global server. For the next round of communication ($r$), a global model sends the updated model parameters ($\theta_r^s$) to all clients.

### III. RESULTS & DISCUSSIONS

To demonstrate the robustness and generalizability of a proposed quantum federated learning algorithm, we studied the non-iid data partitioning strategy in the experiments. We first define the problem studied in this paper and then delve into the details of our results.

### A. Study population

In this paper, we studied three different datasets. 1. Heart disease dataset (HDD) [17]: It comprises four well-known heart disease databases: Cleveland, Hungarian, Switzerland, and Long Beach VA. Originally, this dataset contained 76 attributes, but only a subset of 14 attributes was utilized in the analysis, (13 as predictors and 1 as outcome) [17]. The outcome is an integer valued from 0 (no presence) to 4. For experiments, we have considered class 0 (no heart disease (NHD)) versus class 1, 2, 3, 4 together (heart disease (HD)). 2. Pima Indian diabetes dataset (PIDD) [17]: The National Institute of Diabetes and Digestive and Kidney Diseases released the PIDD dataset. It consists of 9 attributes (8 predictors and 1 class label). It is the representation of 8 characteristics of 768 women having more than 21 years [17]. The objective of the dataset is to diagnostically predict whether or not a patient has diabetes, based on certain diagnostic measurements included in the dataset. 3. We employed the Wisconsin Breast Cancer Dataset (WBCD) from the UCI Machine Learning repository [17]. It has 699 instances that are classified as benign and malignant. There are 569 data points in the dataset: 212 as Malignant, and 357 as Benign. Table I. Summary of datasets detailing the train-test-split and subject counts for three distinct datasets: Heart Disease, Diabetes, and Breast Cancer Wisconsin. Each dataset is categorized by its respective target classes, such as Heart Disease (with and without heart disease), Diabetes (diabetic and non-diabetic),

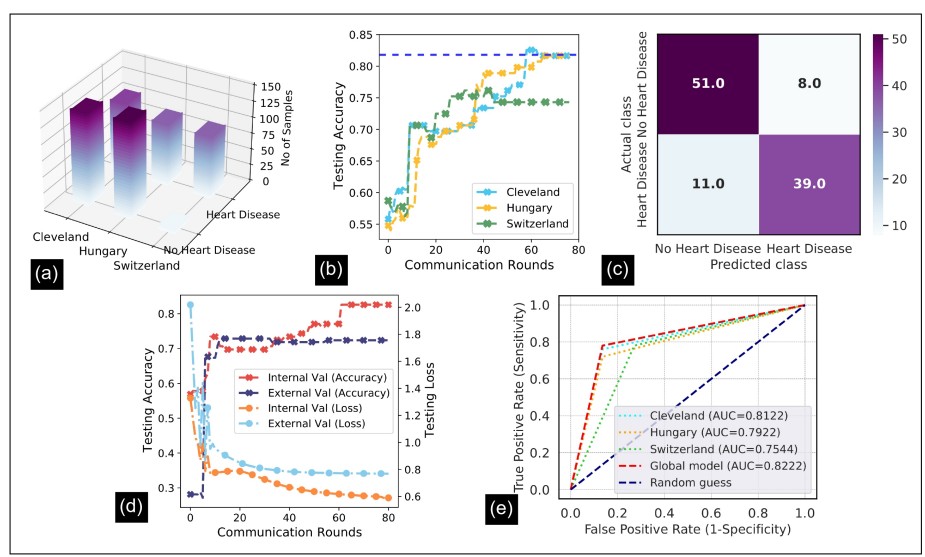

Fig. 4. **Internal and external validation of FedVQC on the Heart Disease (HD) dataset**. (a) 3D plot: Bar plot visualizing the binary classification results for Heart Disease and No Heart Disease labels, accompanied by the distribution of samples from Cleveland, Hungary, and Switzerland Hospitals contributing to each classification category. The blue dotted horizontal line indicates the target accuracy, achieved through training on the entire dataset (i.e., without federated learning). (b) Testing accuracy curves for the three locally trained hospitals on the HD dataset. (c) Confusion Matrix: The diagram depicts the classification outcomes of Heart Disease and No Heart Disease instances within the testing dataset of a FedVQC framework. (d) The global FedVQC model significantly outperforms the Internal validation dataset as compared to the External validation dataset, achieving a testing accuracy of 82% along with a smoother convergence. (e) The AUC-ROC curve plot illustrates the classification performance of heart disease and no heart disease datasets across three datasets. The highest AUC is observed with the global FedVQC model (AUC= 0.822%)

TABLE I
SUMMARY OF DATASETS

|  | Heart Disease dataset [17] | | Diabetes dataset [17] | | Breast Cancer Wisconsin dataset [17] | |
|---|---|---|---|---|---|---|
| Train-Test-Split | Heart Disease | No Heart Disease | Diabetic | Non-Diabetic | Malignant | Benign |
| Train subjects | 310 | 301 | 231 | 421 | 177 | 306 |
| Test subjects | 50 | 59 | 37 | 79 | 35 | 51 |

TABLE II
SUMMARY OF VARIATIONAL QUANTUM CIRCUITS

| Datasets | Features | Qubits | Encoding | Depth | Entangle |
|---|---|---|---|---|---|
| Heart Disease | 13 | 13 | Angle | 4 | Linear |
| Diabetes | 8 | 8 | Angle | 5 | Linear |
| Breast Cancer | 32 | 5 | Amplitude | 6 | Strongly |

TABLE III
SUMMARY OF THE DEMOGRAPHICS IN THE HEART DISEASE [17]

|  | Sex | | Age | | Class | |  |
|---|---|---|---|---|---|---|---|
| Hospital database | M | F | Mean | Std | HD | NHD | Total |
| Cleveland | 207 | 97 | 54.3 | 55.5 | 165 | 139 | 304 |
| Hungary | 212 | 81 | 47.8 | 49.0 | 187 | 106 | 293 |
| Switzerland | 113 | 10 | 55.3 | 55.3 | 06 | 107 | 123 |
| VA Long Beach | 194 | 6 | 59.3 | 59.3 | 51 | 149 | 200 |

and Breast Cancer (malignant and benign). Table II provides a summary of variational quantum circuits.

*B. Performance of FedVQC on the Heart disease dataset*

We first tested the diagnosis capacity of VQC to detect the presence of heart disease in patients across three different hospital databases (Cleveland: 304 cases, Hungary: 293 cases, and Switzerland: 123 cases). Table III provides a summary

of demographics for the heart disease dataset sourced from various hospital databases [17]. The table presents information on the distribution of sex (male and female) and age statistics (mean and standard deviation) for each hospital database included in the study.

A total of 720 samples were utilized for classification evaluation. This comprised 15% of each database for internal validation of the global FedVQC model, while the remaining 85% was retained as local training data. The generalizability of FedVQC was validated on the external database. Each set of local training data comprises 13 features, each of which is encoded into a quantum state using angel encoding, utilizing 13 qubits respectively. The data distribution of each hospital is illustrated in Fig 4(a), with the Cleveland hospital having the maximum number of samples and Switzerland having the fewest number of samples (specifically, 6 cases of no heart disease). The internal test dataset contains 109 patient cases (heart disease=50; no heart disease=59). As testing accuracy and loss are widely used to measure the quality of training capability, we consider this metric versus training communication rounds. To efficiently simulate all circuits, we used a QNGD optimizer to refine the VQC parameters of each hospital. We intend to measure the influence of each client in

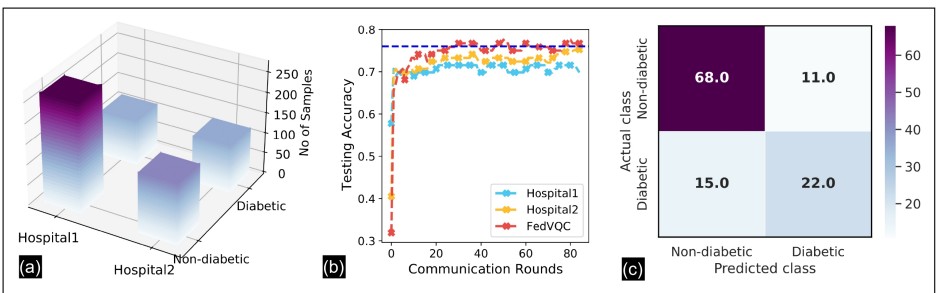

Fig. 5. **Performance of FedVQC on the Diabetes Dataset (DD)**. (a) 3D plot: Bar plot visualizing the binary classification results for diabetic and non-diabetic labels, accompanied by the distribution of samples from Hospital 1 and Hospital 2 contributing to each classification category. (b) Testing Accuracy: Testing accuracy curves for the two locally trained hospitals on the DD dataset. The global FedVQC model significantly outperforms the locally trained models, achieving a testing accuracy of 77.5% along with a smoother convergence. (c) Confusion Matrix: The diagram depicts the classification outcomes of diabetic and non-diabetic instances within the testing dataset of a FedVQC framework.

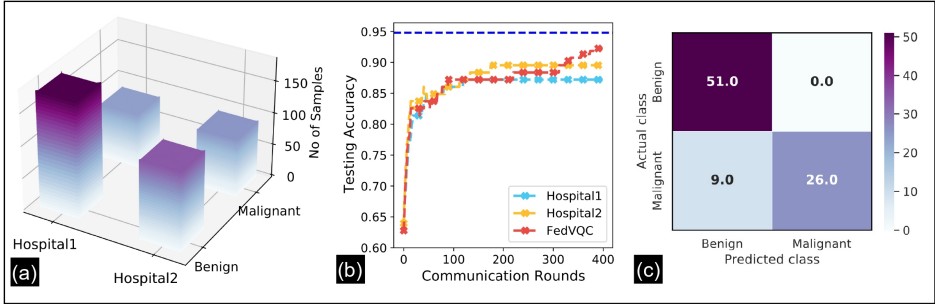

Fig. 6. **Performance of FedVQC on the Breast Cancer Wisconsin (BCW) (Diagnostic) dataset**. (a) 3D plot: Bar plot visualizing the binary classification results for benign and malignant labels, accompanied by the distribution of samples from Hospital 1 and Hospital 2 contributing to each classification category (b) Testing Accuracy: Testing accuracy curves for the two locally trained hospitals on the BCW dataset. The global FedVQC model significantly outperforms the locally trained models, achieving a testing accuracy of 93% along with a smoother convergence. (c) Confusion Matrix: The illustration portrays the classification outcomes of benign and malignant instances within the testing dataset of a global FedVQC.

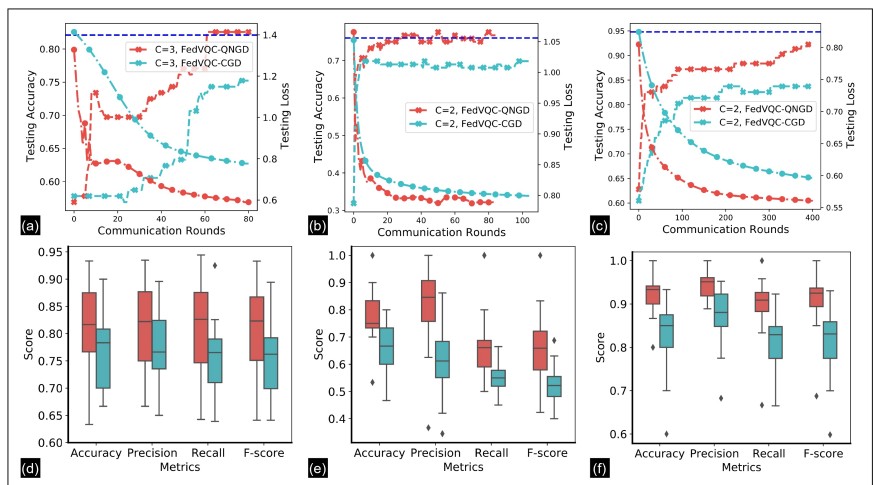

Fig. 7. **Performance and Robustness of FedVQC on different datasets with QNGD and CGD optimizers**. (a)-(c) Testing Accuracy and Loss: A visual representation of the testing accuracy and loss is provided, illustrating the results achieved with the FedVQC model, optimized using both QNGD and CGD optimizers. (d)-(f) Box Plot: The performance is evaluated based on different classification metrics (accuracy, precision, recall, and f-score) on the testing set of the heart disease dataset, diabetes dataset, and breast cancer dataset, respectively.

the training of a global FedVQC model. The testing accuracy of Switzerland is lower due to an imbalanced training sample, in comparison to Cleveland, which has 42% of the samples. For validation, FedVQC first evaluated on the internal testing subsets. To further study how the quantum federated model would generalize to completely unseen centers and patient cohorts, we conducted external validation on the database from Long Beach Medical Center (200 cases). The FedVQC global model outperforms locally trained models with its quicker training convergence and higher test accuracy, as shown in Fig

4(b, d). The confusion matrix and ROC of each local model and global model are depicted in Fig 4(c, e).

## C. Performance of FedVQC on the Diabetes dataset

Next, we evaluated the performance of FedVQC in differentiating between diabetic and non-diabetic patients. The diabetes dataset comprises 8 input features, including age, glucose, insulin, pregnancies, body mass index (BMI), skin thickness, diabetes pedigree function, and blood pressure, along with one binary output feature. Initially, the 8 input features undergo encoding into qubit states via a quantum feature map. It consists of a total 768 patient records. Out of these, the training set consists of 652 (Diabetic: 231; Non-diabetic: 421), and 116 records (Diabetic=37 versus Non-diabetic: 79) were used for testing. The data set is distributed randomly among two hospitals, where hospital1 has the maximum number of samples, i.e., 64% of the training dataset, and hospital2 has 36% of training samples, as shown in Fig. 5(a). The performance of the individual local client model and global FedVQC global model is reflected through testing accuracy against the training communication rounds. Similar to previous results, FedVQC outperformed the local quantum models and achieved higher testing accuracy, after training, as illustrated in Fig 5(b). The blue line indicates the target testing set accuracy. Moreover, the global model requires less than 100 rounds to achieve the target accuracy. We find that FedVQC demonstrates smoother convergence and strong generalization capabilities and attains an impressive testing accuracy of 77.5%, as shown in Fig 5(b). FedVQC classifies diabetic and non-diabetic patients with a specificity of 98.7% and sensitivity of 48%. The confusion matrix of binary label recognition is provided in Fig 5(c).

## D. Performance of FedVQC on the Breast Cancer Wisconsin (Diagnostic) dataset

In this section, we evaluated the performance of FedVQC in distinguishing between benign and malignant cells, representing tumor types from the Breast Cancer Wisconsin (Diagnostic) dataset. It comprises 569 patient records with 32 features, divided into training (Malignant=177, Benign=306) and testing (Malignant=35, Benign=51) sets. To efficiently simulate a quantum circuit, each patient record with 32 features is encoded into a quantum state using amplitude encoding with 5 qubits, followed by a dense quantum circuit, shown in Fig 3(b). Fig 6(a) illustrates the distribution of benign and malignant records across two hospitals. Notably, hospital1 accounts for around 58% of the total samples, while hospital2 represents 42% of the training dataset. The performance of each client and global FedVQC is evaluated after each communication round. As a consequence of the data imbalance, there is an improvement in convergence after 250 communication rounds. Despite the challenges posed by inadequate and unbalanced hospital training data, together with 32 features, the FedVQC model achieved an accuracy of 0.9258, with a specificity of 100% and a sensitivity of 74.28%. FedVQC model achieved a higher true-positive rate

TABLE IV
PERFORMANCE COMPARISON OF FEDVQC WITH CGD AND QNGD

| Metrics | Heart Disease | | Diabetes dataset | | Breast Cancer | |
|---|---|---|---|---|---|---|
| | CGD | QNGD | CGD | QNGD | CGD | QNGD |
| CRounds | 80 | 62 | 100 | 40 | 400 | 380 |
| Accuracy | 78.5 | 82.5 | 68.1 | 77.5 | 85.5 | 89.5 |
| Precision | 76.2 | 82.7 | 61.2 | 66.6 | 88.8 | 1.0 |
| Recall | 76.1 | 82.6 | 53.4 | 59.4 | 83.2 | 85.0 |
| F-score | 76.2 | 82.5 | 51.2 | 66.6 | 83.4 | 91.8 |

and effectively mitigated false positives. Fig 6(c) presents the confusion matrix of the FedVQC model after 400 rounds of communication.

## E. Comparison

We compare our proposed QFL framework consisting VQC algorithm with a classical GD and quantum NGD optimizer. In all three medical datasets, our experimental results indicate that FedVQC with QNGD outperformed FedVQC with a CGD optimizer and significantly reduced the number of communication rounds, as shown in Fig 7(a-c). The box plots illustrate the performance metrics, including testing accuracy, precision, recall, and f-score of federated VQC across all datasets using CGD and QNGD optimizers. The blue line indicates the target testing set accuracy achieved by classical SVM. We address the communication rounds required to reach the target testing set accuracy. FedVQC-QNGD improves accuracy and convergence speed significantly and consistently compared with the FedVQC-CGD approach.FedVQC-QNGD achieved better convergence than the FedVQC-CGD method, resulting in higher precision, recall, and F-score, as depicted in Fig 7(d-f), clearly indicating a significant reduction in communication rounds. FedVQC with quantum optimizer can reduce the communication overhead cost and maintain the baseline performance of binary classifications on the three medical datasets. Therefore, the optimizer selection plays a critical role in optimizing the local client models in federated settings. In Table IV, we compare the mean performance of CGD and QNGD with FedVQC.

Our FedVQC model with quantum optimizer successfully discovered useful mechanism insights to guarantee sorting accuracies. Fig 8(a-c) presents the top-hit feature importance of the Heart disease, diabetes, and breast cancer dataset. These visualizations elucidate the significant features identified through comprehensive analysis, offering insights into their collective relevance and potential impact on classification and prediction tasks. Next, we visualize the loss landscape of FedVQC-QNGD and FedVQC-CGD in Fig 9(a, b), showcasing a local loss function for the classification task of distinguishing between diabetic and non-diabetic instances. The 3D landscape plot showcases the optimization landscape of the FedVQC loss function for two parameters ($\theta_1$ and $\theta_2$), highlighting QNGD's superior performance, nearing perfection, compared to the classical optimizer in federated settings. Classical GD represents the traditional approach, while QNGD leverages quantum techniques for optimization. The bulk of zero eigen-

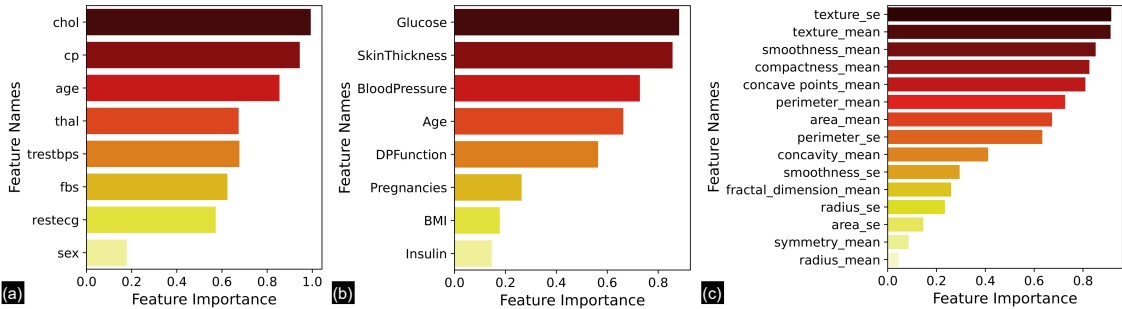

Fig. 8. **Representing the Feature importance in descending order as determined by global model FedVQC**. Representation of top-hit feature importance of (a) Heart disease (b) Diabetes and (c) Breast cancer dataset.

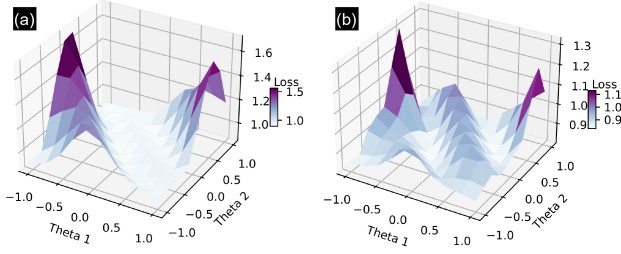

Fig. 9. **Visualizing the 3D plot: Classical vs Quantum optimization**. The loss landscape of CGD and QNGD is demonstrated with a local loss function for the classification task distinguishing between diabetic and non-diabetic instances.

values shows a flat direction of the $\theta$ in the loss landscape of CGD, whereas QNGD optimizer generalizes well.

## IV. CONCLUSION

In this work, we have demonstrated: (a) variational quantum circuits can successfully diagnose diseases within a federated environment. and (b) The choice of optimizer plays a crucial role in achieving effective generalization across clients or hospitals. FedVQC would provide robust/scalable classification for detecting various diseases, validated internally and externally. Their ability to leverage quantum computing capabilities alongside distributed data sources enables accurate and privacy-preserving disease diagnosis across multiple healthcare institutions. We demonstrated the convergence of QNGD in federated settings under different data distributions and presented extensive empirical results. Our findings show the superiority of QNGD over classical CGD in various scenarios, without the need for any tuning. This benefit is crucial, particularly as the increase in communication costs poses challenges for many practical FL applications, especially those involving clients with limited network bandwidth.

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
