# OpenReview forum: "Communication-efficient Quantum Federated Learning Optimization for Multi-Center Healthcare Data"
_IEEE.org/EMBS/BHI/2024/Conference — IEEE BHI'24_

### Official Review · Reviewer_16Yj · 2024-08-07
**Quantum Federated Learning Model for Multi-Center Healthcare Data**

**Overall Rating:** 6
**Confidence:** 4

**Other Quality Metrics:**

- *Clarity of writing: Good*
- *Clinical Significance: Fair*
- *Methodological Novelty: Excellent*
- *Experiments and Results: Good*

**Questions For The Authors:**

- Page 1: “Till now, several approaches have been proposed and shown great potential to revolutionize various sectors, including healthcare, finance, chemistry, cybersecurity, optimization, and many more [4], [5]”

    → How could Quantum Machine Learning revolutionize these areas? For example, can it provide much more powerful model accuracy? Or can it offer a more cost-efficient (energy-friendly) solution?

- Page 2: “application of quantum federated machine learning to handle privacy-sensitive clinical data [15].”

    → How was quantum machine learning utilized here? How was the performance in the previous work, or how is it different from this work?

- Page 3: “a quantum federated model is introduced to address the reluctance of healthcare entities to share data directly. This model not only prioritizes data privacy but also fosters collaborative analysis and insights, thereby presenting a holistic solution to the challenges faced in the healthcare sector.”

    → Similar comment here. How and why can a quantum model help with data privacy issues? Why can’t optimized federated learning with classical machine learning be used given the challenges in quantum computation?

- Page 3: “Section 4 provides details on the dataset,” → This may be referring to Section 3
- Page 4 Section A: “where θr is the parameter vector at round rm η is the learning rate” → round rm might be typo?
- Page 5 Table 1: It would be helpful if the table included how many instances came from each source within the train or test split for each dataset
- Page 5 Section B: “A total of 720 samples were utilized for classification evaluation”

    → How many samples were there for each subject? What were the samples? Were they measurements of each subject across different time points?

- Figure 4(e) may be confusing to read: Describe or put a legend indicating which trace represents accuracy vs. loss
- In Figure 4, there is no figure (d) showing global FedVQC model performance
- In Figures 4, 5, 6 (b), what is the blue dotted horizontal line?
- Page 7 Section E: The text says figure 8, but it seems to be referring to figure 7

**Strengths:**

- The paper is well-structured. The overall architecture of the system is clearly described with detailed illustrations.
- The method part is clearly explained. Although the topic is complex, it is well described in the methods section and easy to follow.
- The topic is novel and interesting and could potentially address data scarcity and data privacy issues in machine learning for healthcare.
- The experiments and results are clearly presented and convincing.

**Summary Of The Paper:**

The paper presents a Quantum Federated Learning (QFL) framework that trains models using quantum computers in a federated learning approach. This allows different medical centers to collaboratively train a global model without sharing their data. In the proposed QFL framework, the paper introduces the quantum natural gradient descent (QNGD) technique, which enhances the performance of the quantum machine learning model compared to the classical gradient descent approach. Overall, the paper demonstrates the feasibility of using the QFL framework for various classification tasks on clinical datasets without privacy concerns and excessive costly communication rounds across distributed medical centers. The paper also shows improved performance using the QNGD technique proposed in the work.

**Weaknesses:**

- Quantum machine learning is a very novel and experimental area. While the motivation behind federated learning in this paper was clear, the motivation for quantum machine learning was not as clear. The paper describes methods to handle quantum machine learning algorithms in detail, but it should include more discussion on why quantum computing could be beneficial in this domain, given the challenges in quantum machine learning. The motivation behind quantum computing (reduced communication rounds and privacy protection) wasn’t clear to me. The work should emphasize more how this could be more beneficial than traditional federated learning algorithms.
- In the results section, there are some missing figures, missing descriptions (e.g., what are the samples, are they the measurements of each subject in different time point?), and some typos that need to be confirmed and clarified. (See Questions in below section for more details)
- In the comparison, is there any performance data for classical federated machine learning models (not on quantum computers)? How does the performance of the quantum machine learning model in this work compare to classical models? Is there any cost gain from the reduced number of communications or more efficient computing?

---

> ### Author Rebuttal · Authors · 2024-09-02
>
> **A1.** Thank you for your insightful comment. QML holds significant promise for revolutionizing various sectors due to its unique computational advantages. **Healthcare:** Enhances drug discovery by analyzing chemical spaces and predicting molecular interactions more accurately [1, 2]. **Finance:** Improves optimization tasks like portfolio management with exponential speed [3]. **Chemistry:** Provides accurate simulations of electronic structures, accelerating material and chemical development  [4, 5]. **Cybersecurity:**  Quantum algorithms, including quantum key distribution (QKD), offer theoretically unbreakable encryption to protect sensitive information [6, 7]. Overall, QML is poised to deliver significant advancements in model accuracy and cost-efficiency.  Its ability to handle complex computations and large datasets more effectively than classical methods positions QML as a powerful tool for addressing some of the most challenging problems across diverse fields.
>
> **A2.** In the earlier work referenced [15], the framework proposed a quantum convolutional neural network (QCNN) with classical optimization specifically designed for medical image data, such as COVID-19 and kidney CT scans. The quantum simulation experiments achieved performance levels on par with well-known classical CNN models for pneumonia and CT-kidney datasets, despite using only a few hundred model parameters. In contrast, our current work advances the use of variational quantum approach in federated learning, specifically targeting different tabular medical datasets. We conduct a comparative analysis of quantum versus classical optimization strategies to evaluate their relative effectiveness and performance in this context.
>
> **A3.** Thank you for your insightful comment. While optimized federated learning with classical machine learning does offer privacy benefits, a quantum federated model has the potential to enhance data privacy through quantum cryptographic techniques, such as quantum key distribution and homomorphic encryption, which are inherently more secure due to principles of quantum mechanics. Moreover, the efficient training of quantum circuits and their capability to differentiate and process complex, large-scale medical datasets in high-dimensional Hilbert space provide advantages in scenarios where classical machine learning models may struggle. Although quantum computing is still in its early stages, its integration into federated learning offers a forward-looking approach that could potentially surpass the capabilities of classical methods in the long term.
>
> **A4.** Thank you for pointing that out. It should refer to Section 3. We have updated the text accordingly.
>
> **A5.** Thank you for catching that typo. It should indeed be 'round r' instead of 'round rm.' The correction has been made.
>
> **A6.** Table 1 has been updated to clearly show the number of samples in each class within the training set and test set.
>
> **A7.** The dataset consists of 720 samples collected from four hospitals, each contributing a different number of samples related to heart disease classification. Here's the breakdown: Cleveland hospital (#304): Heart disease (165) and No Heart disease (139), Hungary hospital (#293): Heart disease (187) and No Heart disease (106), Switzerland hospital (#113): Heart disease (06) and No Heart disease (107), VA long beach hospital (#194): Heart disease (45) and No Heart disease (149).  The samples are not measurements taken across different time points for each subject; rather, they are individual records representing a single measurement or observation per patient. Each record is used to classify whether the patient has heart disease or not, based on the different features.
>
> **A8.** We understand that it might be confusing to read. In the revised paper, we have added a legend to clearly differentiate between the traces representing accuracy and loss. We hope this improves the clarity of the figure.
>
> **A9.** Thank you for pointing out the mistake in Figure 4. We have corrected the numbering error in the revised version of the paper, and the figure now accurately reflects the global FedVQC model performance.
>
> **A10.** The blue dotted horizontal line indicates the target accuracy, achieved through training on the entire dataset (i.e., without federated learning). We have mentioned it in the figure captions.
>
> **A11.** Thank you for bringing it to our attention. It appears there was a typographical error in Page 7, Section E, where "Figure 8" was mistakenly mentioned instead of "Figure 7." We have corrected this error in the revised manuscript.
>
> We would like to express our profound gratitude for your time and insightful comments. We hope our responses address your concerns and that you might reconsider the score. Thank you again for your time in reviewing our manuscript.

---

### Official Review · Reviewer_qAtf · 2024-08-08
**This paper proposes a Quantum Federated Learning (QFL) framework that integrates variational quantum circuits (VQCs) with federated learning to enhance the efficiency of multi-center healthcare data analysis.**

**Overall Rating:** 7
**Confidence:** 3

**Other Quality Metrics:**

Clarity of Writing: Good
Clinical Significance: Great
Methodological Novelty: Good
Experiments and Results: Good

**Questions For The Authors:**

1. How do you plan to validate the generalizability of your approach on more diverse and larger healthcare datasets, especially those representing varied patient populations?
2. What are the specific hardware and computational requirements for implementing your QFL framework, and how do these compare with the resources typically available in healthcare institutions?

**Strengths:**

The integration of quantum computing with federated learning, as proposed in this paper, is a novel approach that leverages the strengths of both fields. This method addresses the challenges of data privacy and distributed learning in healthcare.
Although research in this area is still emerging, similar studies have been conducted, such as "Federated Quantum Machine Learning" by Chen and Yoo (2021, https://www.mdpi.com/1099-4300/23/4/460), which explores quantum neural networks in federated learning settings to enhance privacy and reduce communication costs. Additionally, the work by Huang et al. (2022, https://ieeexplore.ieee.org/document/9763352) on Quantum Federated Learning with decentralized data further demonstrates the potential of this integrated approach.
While the integration of quantum computing with federated learning is still in its early stages, it remains a relatively new and emerging field. The method proposed in this paper contributes to this emerging field by introducing specific optimizations (Quantum Natural Gradient Descent) and applying them to real-world healthcare data, which adds to the practical relevance.
More importantly, the use of QNGD significantly reduces communication rounds, which is a critical improvement over traditional federated learning methods. This is especially beneficial in healthcare settings where data transfer can be a bottleneck due to privacy concerns and network limitations.

**Summary Of The Paper:**

The paper presents a Quantum Federated Learning (QFL) framework designed to address the challenges of privacy and data sharing in multi-center healthcare studies.
The framework leverages Variational Quantum Circuits (VQCs) and introduces Quantum Natural Gradient Descent (QNGD) to optimize the training process, reducing the number of communication rounds between the central server and distributed clients. Specifically, QNGD outperformed classical GD by reducing communication rounds by a range of 5% to 60%.
The study validates this approach using three healthcare datasets—heart disease, diabetes, and breast cancer—demonstrating that the quantum optimization method outperforms classical gradient descent in terms of accuracy and convergence speed.

**Weaknesses:**

The integration of quantum computing into federated learning, while innovative, adds a layer of complexity that may be challenging to implement in practice, particularly for institutions without access to quantum computing resources. The paper could benefit from a discussion on the practical deployment of this technology in real-world healthcare settings.

---

> ### Author Rebuttal · Authors · 2024-09-02
>
> **Q1. How do you plan to validate the generalizability of your approach on more diverse and larger healthcare datasets, especially those representing varied patient populations?**
>
> **A1**. We thank the Reviewer for your insightful question. We implemented the proposed framework using UCI medical datasets, which were distributed across various clients and hospitals. To ensure the generalizability of our approach across diverse healthcare settings, we plan to evaluate it on a broader spectrum of datasets. This will include datasets representing different diseases, sourced from various institutions and geographical regions. By incorporating datasets with diverse demographic characteristics and clinical conditions, we aim to thoroughly assess the adaptability and robustness of our model. Additionally, we will test our approach on well-known publicly available datasets, such as the MIMIC-III Critical Care Database (which consists of intensive care unit (ICU) stays for 100 patients), the SEER Cancer Registry (foe evaluating thyroid cancer prognosis of patients), and the Cancer Genome Atlas, to evaluate its performance across varied patient populations.
>
> **Q2. What are the specific hardware and computational requirements for implementing your QFL framework, and how do these compare with the resources typically available in healthcare institutions?**
>
> **A2**. Thank you for raising an important question about the hardware and computational requirements for our Quantum Federated Learning (QFL) framework. Our proposed framework leverages Variational Quantum Classifiers (VQC), Quantum Natural Gradient Descent (QNGD) and Gradient Descent (GD) optimizers. The specific hardware and computational requirements for implementing this framework are as follows:
>
> Quantum Processor: The VQC and QNGD algorithms require access to quantum processors capable of executing variational quantum circuits. As of now, these resources are not typically available directly within healthcare institutions (except Cleveland Clinic, Ohio, US). However, these processors are available through cloud-based quantum computing platforms such as IBM Quantum, Google Quantum AI, Microsoft Azure Quantum, and many more. Healthcare institutions may need to adapt their infrastructure or collaborate with these technology providers to bridge the gap between quantum computing requirements and their current capabilities.
>
> Simulation:  To handle large-scale data processing and optimization tasks, quantum simulations are performed on high-performance computing clusters (i.e. A100 GPU server), utilized in our simulations. While healthcare institutions often have access to high-performance computing resources, such as GPUs or TPUs, the scale of these resources can vary significantly. The specific hardware needs can be met through cloud computing platforms (Google, Amazon, and NVIDIA). Healthcare institutions can utilize these cloud services to access the necessary computational power, albeit with considerations for cost and integration into their existing IT infrastructures.
>
> We would like to express our profound gratitude for your time and insightful comments. We hope the above answers clarify your concerns, and if so, we hope you could consider reflecting on the score. Thank you again for your time in reviewing our manuscript.

---

### Official Review · Reviewer_SjW5 · 2024-08-11
**Review of the paper "Communication-efficient Quantum Federated Learning Optimization for Multi-Center Healthcare Data"**

**Overall Rating:** 6
**Confidence:** 3

**Other Quality Metrics:**

(a) Clarity of writing: good
(b) Clinical Significance: fair
(c) Methodological Novelty: good
(d) Experiments and Results: good

**Questions For The Authors:**

It should be specified in the abstract that the results were only obtained on UCI datasets.

**Strengths:**

The authors proposed a quantum federated learning framework (QFL) based on variational quantum algorithms and demonstrated its learning capability on popular UCI medical machine learning datasets.

**Summary Of The Paper:**

The authors propose a communication-efficient Quantum Federated Learning (QFL) framework based on a variational circuit that enables clients to efficiently train and transmit quantum model parameters, so reducing communication rounds significantly and enhancing QFL performance using quantum natural gradient descent optimization.

**Weaknesses:**

Only UCI datasets have been used.
No validation on real-world data is provided.

---

> ### Author Rebuttal · Authors · 2024-09-02
>
> **Q1. It should be specified in the abstract that the results were only obtained on UCI datasets.**
>
> **A1**. We would like to thank the Reviewer for their suggestions. We have updated the abstract to specify that the results were obtained solely on UCI datasets. Additionally, we have noted that future work, will include the incorporation of real-world medical datasets to further validate the model. We hope these updates address your concerns and would greatly appreciate it if you consider reflecting on the score. Thank you again for your valuable time and input in reviewing our work.

---

### Decision · Program_Chairs · 2024-09-23

Accept